# Robust-Adaptive Control of Linear Systems: beyond Quadratic Costs

**Edouard Leurent**
Univ. Lille, Inria, CNRS, Centrale Lille, UMR 9189 – CRIStAL, Renault
F-59000 Lille, France
edouard.leurent@inria.fr

**Denis Efimov**
Univ. Lille, Inria, CNRS, Centrale Lille, UMR 9189 – CRIStAL
F-59000 Lille, France
denis.efimov@inria.fr

**Odalric-Ambrym Maillard**
Univ. Lille, Inria, CNRS, Centrale Lille, UMR 9189 – CRIStAL
F-59000 Lille, France
odalric.maillard@inria.fr

## Abstract

We consider the problem of robust and adaptive model predictive control (MPC) of
a linear system, with unknown parameters that are learned along the way (adaptive),
in a critical setting where failures must be prevented (robust). This problem has
been studied from different perspectives by different communities. However, the
existing theory deals only with the case of quadratic costs (the LQ problem), which
limits applications to stabilisation and tracking tasks only. In order to handle
more general (non-convex) costs that naturally arise in many practical problems,
we carefully select and bring together several tools from different communities,
namely non-asymptotic linear regression, recent results in interval prediction, and
tree-based planning. Combining and adapting the theoretical guarantees at each
layer is non trivial, and we provide the first end-to-end suboptimality analysis for
this setting. Interestingly, our analysis naturally adapts to handle many models and
combines with a data-driven robust model selection strategy, which enables to relax
the modelling assumptions. Last, we strive to preserve tractability at any stage of
the method, that we illustrate on two challenging simulated environments.[1]

## 1   Introduction

Despite the recent successes of Reinforcement Learning [e.g. 32, 38], it has hardly been applied in
real industrial issues. This could be attributed to two undesirable properties which limit its practical
applications. First, it depends on a tremendous amount of interaction data that cannot always be
simulated. This issue can be alleviated by model-based methods – which we consider in this work –
that often benefit from better sample efficiencies than their model-free counterparts. Second, it relies
on trial-and-error and random exploration. In order to overcome these shortages, and motivated by
the path planning problem for a self-driving car, in this paper we consider the problem of controlling
an unknown linear system $x(t)$ so as to maximise an *arbitrary* bounded reward function $R$, in a

critical setting where mistakes are costly and must be avoided at all times. This choice of rich reward space is crucial to have sufficient flexibility to model non-convex and non-smooth functions that naturally arise in many practical problems involving combinatorial optimisation, branching decisions, etc., while quadratic costs are mostly suited for tracking a fixed reference trajectory [e.g. 23]. Since experiencing failures is out of question, the only way to prevent them from the outset is to rely on some sort of prior knowledge. In this work, we assume that the system dynamics are partially known, in the form of a linear differential equation with unknown parameters and inputs. More precisely, we consider a linear system with state $x \in \mathbb{R}^p$, acted on by controls $u \in \mathbb{R}^q$ and disturbances $\omega \in \mathbb{R}^r$, and following dynamics in the form:

$$\dot{x}(t) = A(\theta)x(t) + Bu(t) + D\omega(t), \ t \geq 0, \tag{1}$$

where the parameter vector $\theta$ in the state matrix $A(\theta) \in \mathbb{R}^{p \times p}$ belongs to a compact set $\Theta \subset \mathbb{R}^d$. The control matrix $B \in \mathbb{R}^{p \times q}$ and disturbance matrix $D \in \mathbb{R}^{p \times r}$ are known. We also assume having access to the observation of $x(t)$ and to a noisy measurement of $\dot{x}(t)$ in the form $y(t) = \dot{x}(t) + C\nu(t)$, where $\nu(t) \in \mathbb{R}^s$ is a measurement noise and $C \in \mathbb{R}^{p \times s}$ is known. Assumptions over the disturbance $\omega$ and noise $\nu$ will be detailed further, and we denote $\eta(t) = C\nu(t) + D\omega(t)$. We argue that this structure assumption is realistic given that most industrial applications to date have been relying on physical models to describe their processes and well-engineered controllers to operate them, rather than machine learning. Our framework relaxes this modelling effort by allowing some *structured uncertainty* around the nominal model. We adopt a data-driven scheme to estimate the parameters more accurately as we interact with the true system. Many model-based reinforcement learning algorithms rely on the estimated dynamics to derive the corresponding optimal controls [e.g. 24, 28], but suffer from *model bias*: they ignore the error between the learned and true dynamics, which can dramatically degrade control performances [37].

To address this issue, we turn to the framework of *robust* decision-making: instead of merely considering a point estimate of the dynamics, for any $N \in \mathbb{N}$, we build an entire *confidence region* $\mathcal{C}_{N,\delta} \subset \Theta$, illustrated in Figure 1, that contains the true dynamics parameter with high probability:

$$\mathbb{P}\left(\theta \in \mathcal{C}_{N,\delta}\right) \geq 1 - \delta, \tag{2}$$

where $\delta \in (0,1)$. In Section 2, having observed a history $\mathcal{D}_N = \{(x_n, y_n, u_n)\}_{n \in [N]}$ of transitions, our first contribution extends the work of Abbasi-Yadkori et al. [2] who provide a confidence ellipsoid for the least-square estimator to our setting of feature matrices, rather than feature vectors.

The *robust control objective* $V^r$ [8, 9, 18] aims to maximise the worst-case outcome with respect to this confidence region $\mathcal{C}_{N,\delta}$:

$$\sup_{\mathbf{u} \in (\mathbb{R}^q)^{\mathbb{N}}} V^r(\mathbf{u}), \qquad \text{where} \qquad V^r(\mathbf{u}) \stackrel{def}{=} \inf_{\substack{\theta \in \mathcal{C}_{N,\delta} \\ \omega \in [\underline{\omega}, \overline{\omega}]^{\mathbb{R}}}} \left[ \sum_{n=N+1}^{\infty} \gamma^n R(x_n(\mathbf{u}, \omega)) \right], \tag{3}$$

$\gamma \in (0,1)$ is a discount factor, and $x_n(\mathbf{u}, \boldsymbol{\omega})$ is the state reached at step $n$ under controls $\mathbf{u}$ and disturbances $\omega(t)$ within the given admissible bounds $[\underline{\omega}(t), \overline{\omega}(t)]$. Maximin problems such as (3) are notoriously hard if the reward $R$ has a simple form. However, without a restriction on the shape of functions $R$, we cannot hope to derive an explicit solution. In our second contribution, we propose a robust MPC algorithm for solving (3) numerically. In Section 3, we leverage recent results from the uncertain system simulation literature to derive an *interval predictor* $[\underline{x}(t), \overline{x}(t)]$ for the system (1), illustrated in Figure 2. For any $N \in \mathbb{N}$, this predictor takes the information on the current state $x_N$, the confidence region $\mathcal{C}_{N,\delta}$, planned control sequence $\mathbf{u}$ and admissible disturbance bounds $[\underline{\omega}(t), \overline{\omega}(t)]$; and must verify the *inclusion property*:

$$\underline{x}(t) \leq x(t) \leq \overline{x}(t), \forall t \geq t_N. \tag{4}$$

Since $R$ is generic, potentially non-smooth and non-convex, solving the optimal – not to mention the robust – control objective is intractable. In Section 4, facing a sequential decision problem with continuous states, we turn to the literature of tree-based planning algorithms. Although there exist works addressing continuous actions [10, 41], we resort to a first approximation and discretise the continuous decision $(\mathbb{R}^q)^{\mathbb{N}}$ space by adopting a hierarchical control architecture: at each time, the agent can select a high-level *action* $a$ from a finite space $\mathcal{A}$. Each action $a \in \mathcal{A}$ corresponds to the selection of a low-level controller $\pi_a$, that we take affine: $u(t) = \pi_a(x(t)) \stackrel{def}{=} -K_a x(t) + u_a$. For

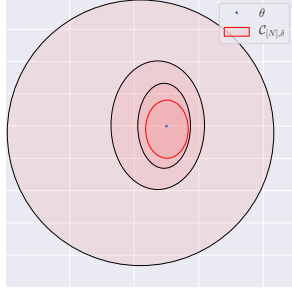
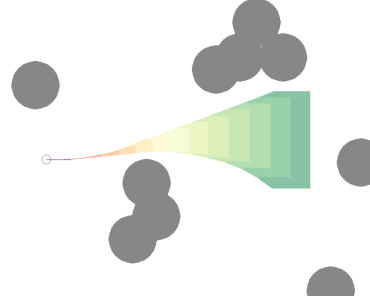

Figure 1: The model estimation procedure, running on the obstacle avoidance problem of Section 6. The confidence region $C_{N,\delta}$ shrinks with the number of samples $N$.

Figure 2: The state prediction procedure running on the obstacle avoidance problem of Section 6. At each time step (red to green), we bound the set of reachable states under model uncertainty (2)

---

**Algorithm 1** Robust Estimation, Prediction and Control

---

**Input:** confidence level $\delta$, structure $(A, \phi)$, reward $R$, $\mathcal{D}_{[0]} \leftarrow \emptyset$, $\mathbf{a}_1 \leftarrow \emptyset$
**for** $N = 0, 1, 2, \ldots$ **do**
    $\mathcal{C}_{N,\delta} \leftarrow$ MODEL ESTIMATION$(\mathcal{D}_N)$. (9)
    **for** each planning step $k \in \{N, \ldots, N + K\} = N + [K]$ **do**
        $[\underline{x}_{k+1}, \overline{x}_{k+1}] \leftarrow$ INTERVAL PREDICTION$(\mathcal{C}_{N,\delta}, \mathbf{a}_k b)$ for each action $b \in \mathcal{A}$. (11)
        $\mathbf{a}_{k+1} \leftarrow$ PESSIMISTIC PLANNING$(\underline{R}_{k+1}([\underline{x}_{k+1}, \overline{x}_{k+1}]))$. (13)
    **end for**
    Execute the recommended control $u_{N+1}$, and add the transition $(x_{N+1}, y_{N+1}, u_{N+1})$ to $\mathcal{D}_{[N+1]}$.
**end for**

---

instance, a tracking a subgoal $x_g$ can be achieved with $\pi_g = K(x_g - x)$. This discretisation induces a suboptimality, but it can be mitigated by diversifying the controller basis. The robust objective (3) becomes $\sup_{\mathbf{a} \in \mathcal{A}^{\mathbb{N}}} V^r(\mathbf{a})$, where $x_n(\mathbf{a}, \omega)$ stems from (1) with $u_n = \pi_{a_n}(x_n)$. However, tree-based planning algorithms are designed for a single known generative model rather than a confidence region for the system dynamics. Our third contribution adapts them to the robust objective (3) by approximating it with a tractable surrogate $\hat{V}^r$ that exploits the interval predictions (4) to define a pessimistic reward. In our main result, we show that the best surrogate performance achieved during planning is guaranteed to be attained on the true system, and provide an upper bound for the approximation gap and suboptimality of our framework in Theorem 3. This is the first result of this kind for maximin control with generic costs to the best of our knowledge. Algorithm 1 shows the full integration of the three procedures of estimation, prediction and control.

In Section 5, our forth contribution extends the proposed framework to consider multiple modelling assumptions, while narrowing uncertainty through data-driven model rejection, and still ensuring safety via robust model-selection during planning.

Finally, in Section 6 we demonstrate the applicability of Algorithm 1 in two numerical experiments: a simple illustrative example and a more challenging simulation for safe autonomous driving.

**Notation** The system dynamics are described in continuous time, but sensing and control happen in discrete time with time-step $dt > 0$. For any variable $z$, we use subscript to refer to these discrete times: $z_n = z(t_n)$ with $t_n = n dt$ and $n \in \mathbb{N}$. We use bold symbols to denote temporal sequences $\mathbf{z} = (z_n)_{n \in \mathbb{N}}$. We denote $z^+ = \max(z, 0)$, $z^- = z^+ - z$, $|z| = z^+ + z^-$ and $[n] = \{1, \ldots, n\}$.

## 1.1 Related Work

The control of uncertain systems is a long-standing problem, to which a vast body of literature is dedicated. Existing work is mostly concerned with the problem of *stabilisation* around a fixed reference state or trajectory, including approaches such as $\mathcal{H}_\infty$ control [7], sliding-mode control [31] or system-level synthesis [11, 12]. This paper fits in the popular MPC framework, for which

adaptive data-driven schemes have been developed to deal with model uncertainty [36, 39, 5], but lack guarantees. The family of tube-MPC algorithms seeks to derive theoretical guarantees of *robust constraint satisfaction*: the state $x$ is constrained in a safe region $\mathbb{X}$ around the origin, often chosen convex [17, 4, 6, 40, 29, 22, 30, 27]. Yet, many tasks cannot be framed as stabilisation problems (e.g. obstacle avoidance) and are better addressed with the minimax control objective, which allows more flexible goal formulations. Minimax control has mostly been studied in two particular instances.

**Finite states** Minimax control of finite Markov Decision Processes with uncertain parameters was studied in [21, 33, 42], who showed that the main results of Dynamic Programming can be extended to their robust counterparts only when the dynamics ambiguity set verifies a certain rectangularity property. Since we consider continuous states, these methods do not apply.

**Linear dynamics and quadratic costs** Several approaches have been proposed for cumulative regret minimisation in the LQ problem. In the *Optimism in the Face of Uncertainty* paradigm, the best possible dynamics within a high-confidence region is selected under a controllability constraint, to compute the corresponding optimal control in closed-form by solving a Riccati equation. The results of [1, 20, 16] show that this procedure achieves a $\widetilde{\mathcal{O}}\left(N^{1/2}\right)$ regret. Posterior sampling algorithms [34, 3] select candidate dynamics randomly instead, and obtain the same result. Other works use noise injection for exploration such as [11, 12]. However, neither optimism nor random exploration fit a critical setting, where ensuring safety requires instead to consider pessimistic outcomes. The work of Dean et al. [11] is close to our setting: after an offline estimation phase, they estimate a suboptimality between a minimax controller and the optimal performance. Our work differs in that it addresses a generic shape cost. Another work of interest is [35] where worst-case generic costs are considered. However, they assume the knowledge of the dynamics, and their rollout-based solution only produces inner-approximations and does not yield any guarantee. In this paper, interval prediction is used to produce oversets, while a near-optimal control is found using a tree-based planning procedure.

## 2  Model Estimation

To derive a confidence region (2) for $\theta$, the functional relationship $A(\theta)$ must be specified.

**Assumption 1** (Structure). *There exists a known feature tensor $\phi \in \mathbb{R}^{d \times p \times p}$ such that for all $\theta \in \Theta$,*

$$A(\theta) = A + \sum_{i=1}^{d} \theta_i \phi_i, \tag{5}$$

*where $A, \phi_1, \ldots, \phi_d \in \mathbb{R}^{p \times p}$ are known. For all $n$, we denote $\Phi_n = [\phi_1 x_n \ldots \phi_d x_n] \in \mathbb{R}^{p \times d}$. We also assume to know a bound $S$ such that $\theta \in [-S, S]^d$.*

We slightly abuse notations and include additional known terms in the measurement signal $y(t) = \dot{x}(t) + C\nu(t) - Ax(t) - Bu(t)$, to obtain a linear regression system $y_n = \Phi_n \theta + \eta_n$.

**Regularised least square** To derive an estimate on $\theta$, we consider the $L_2$-regularised regression problem with weights $\Sigma_p \in \mathbb{R}^{p \times p}$ and $\lambda \in \mathbb{R}_*^+$:

$$\min_{\theta \in \mathbb{R}^d} \sum_{n=1}^{N} \|y_n - \Phi_n \theta\|_{\Sigma_p^{-1}}^2 + \lambda \|\theta\|^2. \tag{6}$$

**Proposition 1** (Regularised solution). *The solution to (6) is*

$$\theta_{N,\lambda} = G_{N,\lambda}^{-1} \sum_{n=1}^{N} \Phi_n^\intercal \Sigma_p^{-1} y_n, \qquad where \quad G_{N,\lambda} = \sum_{n=1}^{N} \Phi_n^\intercal \Sigma_p^{-1} \Phi_n + \lambda I_d \in \mathbb{R}^{d \times d}. \tag{7}$$

Substituting $y_n$ into (7) yields the regression error: $\theta_{N,\lambda} - \theta = G_{N,\lambda}^{-1} \sum_{n=1}^{N} \Phi_n^\intercal \Sigma_p^{-1} \eta_n - \lambda G_{N,\lambda}^{-1} \theta$. To bound this error, we need the noise $\eta_n$ to concentrate.

**Assumption 2** (Noise Model). *We assume that*

1. *at each time $t \geq 0$, the combined noise $\eta(t)$ is an independent sub-Gaussian noise with covariance proxy $\Sigma_p \in \mathbb{R}^{p \times p}$:*

$$\forall u \in \mathbb{R}^p, \ \mathbb{E}\left[\exp\left(u^\intercal \eta(t)\right)\right] \leq \exp\left(\frac{1}{2}u^\intercal \Sigma_p u\right);$$

2. *at each time $t \geq 0$, the disturbance $\omega(t)$ is enclosed by* known *bounds $\underline{\omega}(t) \leq \omega(t) \leq \overline{\omega}(t)$, whose amplitude verifies $\sum_{n=0}^{\infty} \gamma^n C_\omega(t_n) < \infty$, where*

$$C_\omega(t) \overset{def}{=} \sup_{\tau \in [0,t]} \|\overline{\omega}(\tau) - \underline{\omega}(\tau)\|_2.$$

**Theorem 1** (Confidence ellipsoid, a matricial version of Abbasi-Yadkori et al. 2). *Under Assumption 2, it holds with probability at least $1 - \delta$ that*

$$\|\theta_{N,\lambda} - \theta\|_{G_{N,\lambda}} \leq \beta_N(\delta), \quad with \quad \beta_N(\delta) \overset{def}{=} \sqrt{2\ln\left(\frac{\det(G_{N,\lambda})^{1/2}}{\delta \det(\lambda I_d)^{1/2}}\right)} + (\lambda d)^{1/2}S. \quad (8)$$

We convert this confidence ellipsoid $\mathcal{C}_{N,\delta}$ from (8) into a polytope for $A(\theta)$. For simplicity, we present here a simple but coarse strategy: bound the ellipsoid by its enclosing axis-aligned hypercube:

$$A(\theta) \in \left\{ A_N + \sum_{i=1}^{2^d} \alpha_i \Delta A_{N,i} : \alpha \geq 0, \sum_{i=1}^{2^d} \alpha_i = 1 \right\} \quad (9)$$

where $A_N = A(\theta_{N,\lambda})$, $h_i \in \{-1, 1\}^d$, $\Delta A_{N,i} = h_i \sqrt{\frac{\beta_N(\delta)}{\lambda_{\max}(G_{N,\lambda})}}$. A tighter polytope derivation is presented in the Supplementary Material.

## 3 State Prediction

A simple solution to (4) is proposed in [14], where, given bounds $\underline{A} \leq A(\theta) \leq \overline{A}$ from $\mathcal{C}_{N,\delta}$ they use matrix interval arithmetic to derive the predictor:

**Proposition 2** (Simple predictor of Efimov et al. 14). *Assuming that* (2) *is satisfied for the system* (1)*, then the interval predictor following $\underline{x}(t_N) = \overline{x}(t_N) = x(t_N)$ and:*

$$\dot{\underline{x}}(t) = \underline{A}^+ \underline{x}^+(t) - \overline{A}^+ \underline{x}^-(t) - \underline{A}^- \overline{x}^+(t) + \overline{A}^- \overline{x}^-(t) + Bu(t) + D^+\underline{\omega}(t) - D^-\overline{\omega}(t), \quad (10)$$

$$\dot{\overline{x}}(t) = \overline{A}^+ \overline{x}^+(t) - \underline{A}^+ \overline{x}^-(t) - \overline{A}^- \underline{x}^+(t) + \underline{A}^- \underline{x}^-(t) + Bu(t) + D^+\overline{\omega}(t) - D^-\underline{\omega}(t),$$

*ensures the inclusion property* (4) *with confidence level $\delta$.*

However, Leurent et al. [26] showed that this predictor can have unstable dynamics, even for stable systems, which causes a fast build-up of uncertainty. They proposed an enhanced predictor which exploits the polytopic structure (9) to produce more stable predictions, at the price of a requirement:

**Assumption 3.** *There exists an orthogonal matrix $Z \in \mathbb{R}^{p \times p}$ such that $Z^\intercal A_N Z$ is Metzler[2].*

In practice, this assumption is often verified: it is for instance the case whenever $A_N$ is diagonalisable. The similarity transformation of [15] provides a method to compute such $Z$ when the system is observable. To simplify the notation, we will further assume that $Z = I_p$. Denote $\Delta A_+ = \sum_{i=1}^{2^d} \Delta A_{N,i}^+$, $\Delta A_- = \sum_{i=1}^{2^d} \Delta A_{N,i}^-$.

**Proposition 3** (Enhanced predictor of Leurent et al. 26). *Assuming that* (9) *and Assumption 3 are satisfied for the system* (1)*, then the interval predictor following $\underline{x}(t_N) = \overline{x}(t_N) = x(t_N)$ and:*

$$\dot{\underline{x}}(t) = A_N \underline{x}(t) - \Delta A_+ \underline{x}^-(t) - \Delta A_- \overline{x}^+(t) + Bu(t) + D^+\underline{\omega}(t) - D^-\overline{\omega}(t), \quad (11)$$

$$\dot{\overline{x}}(t) = A_N \overline{x}(t) + \Delta A_+ \overline{x}^+(t) + \Delta A_- \underline{x}^-(t) + Bu(t) + D^+\overline{\omega}(t) - D^-\underline{\omega}(t),$$

*ensures the inclusion property* (4) *with confidence level $\delta$.*

Figure 3 compares the performance of the predictors (10) and (11) in a simple example. It suggests to always prefer (11) whenever Assumption 3 is verified, and only fallback to (10) as a last resort.

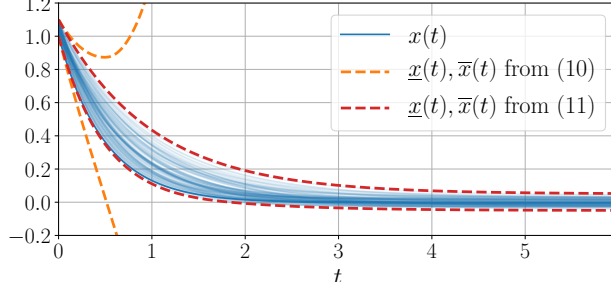

Figure 3: Comparison of (10) and (11) for a simple system $\dot{x}(t) = -\theta x(t) + \omega(t)$, with $\theta \in [1, 2]$ and $\omega(t) \in [-0.05, 0.05]$.

## 4 Robust Control

To evaluate the robust objective $V^r$ (3), we approximate it thanks to the interval prediction $[\underline{x}(t), \overline{x}(t)]$.

**Definition 1** (Surrogate objective). *Let* $\underline{x}_n(\mathbf{u}), \overline{x}_n(\mathbf{u})$ *following the dynamics defined in* (11) *and*

$$\hat{V}^r(\mathbf{u}) \stackrel{def}{=} \sum_{n=N+1}^{\infty} \gamma^n \underline{R}_n(\mathbf{u}) \quad where \quad \underline{R}_n(\mathbf{u}) \stackrel{def}{=} \min_{x \in [\underline{x}_n(\mathbf{u}), \overline{x}_n(\mathbf{u})]} R(x). \tag{12}$$

Such a substitution makes this pessimistic reward $\underline{R}_n$ *not Markovian*, since the worst case is assessed over the whole past trajectory.

**Theorem 2** (Lower bound). *The surrogate objective* (12) *is a lower bound of the objective* (3).

$$\hat{V}^r(\mathbf{u}) \leq V^r(\mathbf{u}) \leq V(\mathbf{u})$$

Consequently, since all our approximations are conservative, if we manage to find a control sequence such that no *"bad event"* (e.g. a collision) happens according to the surrogate objective $\hat{V}^r$, they are *guaranteed* not to happen either when the controls are executed on the true system.

To maximise $\hat{V}^r$, we cannot use DP algorithms since the state space is continuous and the pessimistic rewards are non-Markovian. Rather, we turn to tree-based planning algorithms, which optimise a sequence of actions based on the corresponding sequence of rewards, without requiring Markovity nor state enumeration. In particular, we consider the *Optimistic Planning of Deterministic Systems* (OPD) algorithm [19] tailored for the case when the relationship between actions and rewards is deterministic. Indeed, the stochasticity of the disturbances and measurements is encased in $\hat{V}^r$: given the observations up to time $N$ both the predictor dynamics (11) and the pessimistic rewards in (12) are deterministic. At each planning iteration $k \in [K]$, OPD progressively builds a tree $\mathcal{T}_{k+1}$ by forming upper-bounds $B_a(k)$ over the value of sequences of actions $a$, and expanding[3] the leaf $a_k$ with highest upper-bound:

$$a_k = \arg\max_{a \in \mathcal{L}_k} B_a(k), \quad B_a(k) = \sum_{n=0}^{h(a)-1} \underline{R}_n(a) + \frac{\gamma^{h(a)}}{1-\gamma} \tag{13}$$

where $\mathcal{L}_k$ is the set of leaves of $\mathcal{T}_k$, $h(a)$ is the length of the sequence $a$, and $\underline{R}_n(a)$ the pessimistic reward (12) obtained at time $n$ by following the controls $u_n = \pi_{a_n}(x_n)$.

**Lemma 1** (Planning performance of Hren & Munos 19). *The suboptimality of the* OPD *algorithm* (13) *applied to the surrogate objective* (12) *after $K$ planning iterations is:*

$$\hat{V}^r(a_\star) - \hat{V}^r(a_K) = \mathcal{O}\left(K^{-\frac{\log 1/\gamma}{\log \kappa}}\right);$$

*where* $\kappa \stackrel{def}{=} \limsup_{h \to \infty} \left| \left\{ a \in A^h : \hat{V}^r(a) \geq \hat{V}^r(a^\star) - \frac{\gamma^{h+1}}{1-\gamma} \right\} \right|^{1/h}$ *is a problem-dependent measure of the proportion of near-optimal paths.*

Hence, by using enough computational budget $K$ for planning we can get as close as we want to the optimal surrogate value $\hat{V}^r(a^\star)$, at a polynomial rate. Unfortunately, there exists a gap between $\hat{V}^r$ and the true robust objective $V^r$, which stems from three approximations: (i) the true reachable set was approximated by an enclosing interval in (4); (ii) the time-invariance of the dynamics uncertainty $A(\theta) \in \mathcal{C}_{N,\delta}$ was handled by the interval predictor (11) as if it were a time-varying uncertainty $A(\theta(t)) \in \mathcal{C}_{N,\delta}, \forall t$ ; and (iii) the lower-bound $\sum \min \leq \min \sum$ used to define the surrogate objective (12) is not tight. However, this gap can be bounded under additional assumptions.

**Theorem 3** (Suboptimality bound). *Under two conditions:*

1. *a Lipschitz regularity assumption for the reward function $R$:*

2. *a stability condition: there exist $P > 0, Q_0 \in \mathbb{R}^{p \times p}$, $\rho > 0$, and $N_0 \in \mathbb{N}$ such that*

$$\forall N > N_0, \quad \begin{bmatrix} A_N^\intercal P + P A_N + Q_0 & P|D| \\ |D|^\intercal P & -\rho I_r \end{bmatrix} < 0;$$

*we can bound the suboptimality of Algorithm 1 with planning budget $K$ as:*

$$V(a_\star) - \hat{V}^r(a_K) \leq \underbrace{\Delta_\omega}_{\substack{\text{robustness to} \\ \text{disturbances}}} + \underbrace{\mathcal{O}\left( \frac{\beta_N(\delta)^2}{\lambda_{\min}(G_{N,\lambda})} \right)}_{\text{estimation error}} + \underbrace{\mathcal{O}\left( K^{-\frac{\log 1/\gamma}{\log \kappa}} \right)}_{\text{planning error}}$$

*with probability at least $1 - \delta$, where $V(a)$ is the optimal expected return when executing an action $a \in \mathcal{A}$, $a_\star$ is an optimal action, and $\Delta_\omega$ is a constant which corresponds to an irreducible suboptimality suffered from being robust to instantaneous disturbances $\omega(t)$.*

It is difficult to check the validity of the stability condition 2. since it applies to matrices $A_N$ produced by the algorithm rather than to the system parameters. A stronger but easier to check condition is that the polytope (9) at some iteration becomes included in a set where this property is uniformly satisfied. For instance, if the features are sufficiently excited, the estimation converges to a neighbourhood of the true dynamics $A(\theta)$. This also allows to further bound the input-dependent estimation error term.

**Corollary 1** (Asymptotic near-optimality). *Under an additional persistent excitation (PE) assumption*

$$\exists \underline{\phi}, \overline{\phi} > 0 : \forall n \geq n_0, \quad \underline{\phi}^2 \leq \lambda_{\min}(\Phi_n^\intercal \Sigma_p^{-1} \Phi_n) \leq \overline{\phi}^2, \tag{14}$$

*the stability condition 2. of Theorem 3 can be relaxed to apply to the true system: there exist $P, Q_0, \rho$ such that*

$$\begin{bmatrix} A(\theta)^\intercal P + P A(\theta) + Q_0 & P|D| \\ |D|^\intercal P & -\rho I_r \end{bmatrix} < 0;$$

*and the suboptimality bound takes the more explicit form*

$$V(a_\star) - \hat{V}^r(a_K) \leq \Delta_\omega + \mathcal{O}\left( \frac{\log\left(N^{d/2}/\delta\right)}{N} \right) + \mathcal{O}\left( K^{-\frac{\log 1/\gamma}{\log \kappa}} \right)$$

*which ensures asymptotic near-optimality when $N \to \infty$ and $K \to \infty$.*

## 5 Multi-model Selection

The procedure we developed in Sections 2 to 4 relies on strong modelling assumptions, such as the linear dynamics (1) and Assumption 1. But what if they are wrong?

**Model adequacy** One of the major benefits of using the family of linear models, compared to richer model classes, is that they provide strict conditions allowing to quantify the adequacy of the modelling assumptions to the observations. Given $N - 1$ observations, Section 2 provides a polytopic confidence region (9) that contains $A(\theta)$ with probability at least $1 - \delta$. Since the dynamics are linear, we can propagate this confidence region to the next observation: $y_N$ must belong to the Minkowski sum of a polytope representing model uncertainty $\mathcal{P}(A_0 x_N + B u_N, \Delta A_1 x_N, \dots, \Delta A_{2^d} x_N)$ and a polytope $\mathcal{P}(0_p, \underline{\eta}, \overline{\eta})$ bounding the disturbance and measurement noises. Delos & Teissandier [13] provide a way to test this membership in polynomial time using linear programming. Whenever it is not verified, we can confidently reject the $(A, \phi)$-modelling assumption 1. This enables us to consider a rich set of potential features $\left((A^1, \phi^1), \dots, (A^M, \phi^M)\right)$ rather than relying on a single assumption, and only retain those that are consistent with the observations so far. Then, every remaining hypothesis must be considered during planning.

**Robust selection** We temporarily ignore the parametric uncertainty on $\theta$ to simply consider several candidate dynamics models, which all correspond to different modelling assumptions. We also restrict to deterministic dynamics, which is the case of (11).

**Assumption 4** (Multi-model ambiguity). *The dynamics $f$ lie in a finite set of candidates $(f^m)_{m \in [M]}$.*

We adapt our planning algorithm to balance these concurrent hypotheses in a robust fashion, i.e. maximise a robust objective with discrete ambiguity:

$$V^r = \sup_{a \in \mathcal{A}^{\mathbb{N}}} \min_{m \in [M]} \sum_{n=N+1}^{\infty} \gamma^n R_n^m, \tag{15}$$

where $R_n^m$ is the reward obtained by following the action sequence $a$ up to step $n$ under the dynamics $f^m$. This objective could be optimised in the same way as in Section 4, but this would result in a coarse and lossy approximation. Instead, we exploit the finite uncertainty structure of Assumption 4 to asymptotically recover the true $V^r$ by modifying the OPD algorithm in the following way:

**Definition 2** (Robust UCB). *We replace the upper-bound (13) on sequence values in OPD by:*

$$B_a^r(k) \stackrel{def}{=} \min_{m \in [M]} \sum_{n=0}^{h-1} \gamma^n R_n^m + \frac{\gamma^h}{1 - \gamma}. \tag{16}$$

Note that it is not equivalent to solving each control problem independently and following the action with highest worst-case value, as we show in the Supplementary Material. We analyse the sample complexity of the corresponding robust planning algorithm.

**Proposition 4** (Robust planning performance). *The robust version of OPD (16) enjoys the same regret bound as OPD in Lemma 1, with respect to the multi-model objective (15).*

This result is of independent interest: the solution of a robust objective (15) with discrete ambiguity $f \in \{f^m\}_{m \in [M]}$ can be recovered exactly, asymptotically when the planning budget $K$ goes to infinity, which Robust DP algorithms do not allow. This also contrasts with the results obtained in Section 4 for the robust objective (3) with continuous ambiguity $A(\theta) \in \mathcal{C}_{N,\delta}$, for which OPD only recovers the surrogate approximation $\hat{V}^r$, as discussed in Theorem 3. Note that here the regret depends on the number $K$ of node expansions, but each expansion now requires $M$ times more simulations than in the single-model setting. Finally, the two approaches of Sections 4 and 5 can be merged by using the pessimistic reward (12) in (16).

## 6 Experiments

Videos and code are available at `https://eleurent.github.io/robust-beyond-quadratic/`.

**Obstacle avoidance with unknown friction** We first consider a simple illustrative example, shown in Figure 2: the control of a 2D system moving by means of a force $(u_x, u_y)$ in an medium with anisotropic linear friction with unknown coefficients $(\theta_x, \theta_y)$. The reward encodes the task of navigating to reach a goal state $x_g$ while avoiding collisions with obstacles: $R(x) = \delta(x)/(1 + \|x - x_g\|_2)$ where $\delta(x)$ is 0 whenever $x$ collides with an obstacle, 1 otherwise. The actions $\mathcal{A}$ are constant controls in the up, down, left and right directions. For the reasons mentioned above, no robust baseline applies to our setting. We compare Algorithm 1 to the non-robust adaptive control approach that plans with the estimated dynamics $\theta_{N,\lambda}$, and thus enjoys the same prior knowledge of dynamics structure and reward. This highlights the benefits of the robust formulation solely rather than stemming from algorithm design. We show in Table 1(a) the results of 100 simulations of a single episode: the robust agent performs worse than the nominal agent on average, but manages to ensure safety while the nominal agent collides with obstacles in $4\%$ of simulations. We also compare to a standard model-free approach, DQN, which does not benefit from the prior knowledge on the system dynamics, and is instead trained over multiple episodes. The reported performance is that of the final policy obtained after training for 3000 episodes, during which $897 \pm 64$ collisions occurred ($29.9 \pm 2.1\%$). We study the evolution of the suboptimality $V(x_N) - \sum_{n>N} \gamma^{n-N} R(x_n)$ with respect to the number of samples $N$, by comparing the empirical returns from a state $x_N$ to the value $V(x_N)$ that the agent would get by acting optimally from $x_N$ with knowledge of the dynamics. Although the assumptions

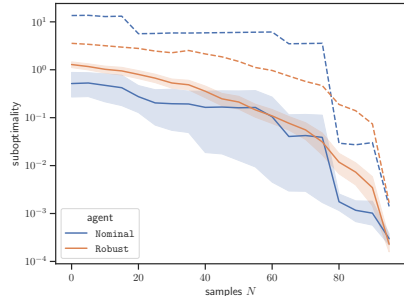

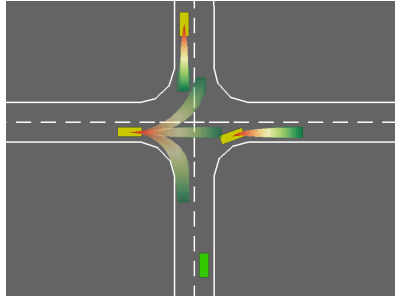

Figure 4: The mean (solid), 95% CI for the mean (shaded) and maximum (dashed) suboptimality with respect to $N$.

Figure 5: The intersection crossing task. Trajectory intervals show behavioural uncertainty for each vehicle, with a multi-model assumption over their route.

Table 1: Frequency of collision, minimum and average return achieved on a single episode, repeated with 100 random seeds. In both tasks, the robust agent performs worse than the nominal agent on average, but manages to ensure safety and attains a better worst-case performance.

| (a) Performances on the obstacle task | | | | (b) Performances on the driving task | | | |
|---|---|---|---|---|---|---|---|
| Performance | failures | min | avg $\pm$ std | Performance | failures | min | avg $\pm$ std |
| Oracle | 0% | 11.6 | $14.2 \pm 1.3$ | Oracle | 0% | 6.9 | $7.4 \pm 0.5$ |
| Nominal | 4% | 2.8 | $\mathbf{13.8} \pm 2.0$ | Nominal 1 | 4% | 5.2 | $\mathbf{7.3} \pm 1.5$ |
| Algorithm 1 | **0%** | **10.4** | $13.0 \pm 1.5$ | Nominal 2 | 33% | 3.5 | $6.4 \pm 0.3$ |
| | | | | Algorithm 1 | **0%** | **6.8** | $7.1 \pm 0.3$ |
| DQN (trained) | 6% | 1.7 | $12.3 \pm 2.5$ | DQN (trained) | 3% | 5.4 | $6.3 \pm 0.6$ |

of Theorem 3 are not satisfied (e.g. non-smooth reward), the mean suboptimality of the robust agent, shown in Figure 4, still decreases polynomially with $N$: Algorithm 1 gets *more efficient* as it is *more confident* while *ensuring safety* at all times. In comparison, the nominal agent enjoys a smaller suboptimality on average, but higher in the worst-case.

**Motion planning for an autonomous vehicle**   We consider the highway-env environment [25] for simulated driving decision problems. An autonomous vehicle with state $\chi_0 \in \mathbb{R}^4$ is approaching an intersection among $V$ other vehicles with states $\chi_i \in \mathbb{R}^4$, resulting in a joint traffic state $x = [\chi_0, \ldots, \chi_V]^\top \in \mathbb{R}^{4V+4}$. These vehicles follow parametrized behaviours $\dot{\chi}_i = f_i(x, \theta_i)$ with unknown parameters $\theta_i \in \mathbb{R}^5$. We appreciate a first advantage of the structure imposed in Assumption 1: the uncertainty space of $\theta$ is $\mathbb{R}^{5V}$. In comparison, the traditional LQ setting where the whole state matrix $A$ is estimated would have resulted in a much larger parameter space $\theta \in \mathbb{R}^{16V^2}$. The system dynamics $f$, which describes the interactions between vehicles, can only be expressed in the form of Assumption 1 given the knowledge of the desired route for each vehicle, with features $\phi$ expressing deviations to the centerline of the followed lane. Since these intentions are unknown to the agent, we adopt the multi-model perspective of Section 5 and consider one model per possible route for every observed vehicle before an intersection. In Table 1(b), we compare Algorithm 1 to a nominal agent planning with two different modelling assumptions: Nominal 1 has access to the true followed route for each vehicle, while Nominal 2 does not and picks the model with minimal prediction error. Again we also compare to a DQN baseline trained over 3000 episodes, causing $1058 \pm 113$ collisions while training $(35 \pm 4\%)$. As before, the robust agent has a higher worst-case performance and avoids collisions at all times, at the price of a decreased average performance..

## Conclusion

We present a framework for the robust estimation, prediction and control of a partially known linear system with generic costs. Leveraging tools from linear regression, interval prediction, and tree-based planning, we guarantee the predicted performance and provide a suboptimality bound. The method applicability is further improved by a multi-model extension and demonstrated on two simulations.

## Broader Impact

The motivation behind this work is to enable the development of Reinforcement Learning solutions for industrial applications, when it has been mainly limited to simulated games so far. In particular, many industries already rely on non-adaptive control systems and could benefit from an increased efficiency, including Oil and Gas, robotics for industrial automation, Data Center cooling, etc. But more often than not, safety-critical constraints proscribe the use of exploration, and industrials are reluctant to turn to learning-based methods that lack accountability. This work addresses these concerns by focusing on risk-averse decisions and by providing worst-case guarantees. Note however that these guarantees are only as good as the validity of the underlying hypotheses, and Assumption 1 in particular should be submitted to a comprehensive validation procedure; otherwise, decisions formed on a wrong basis could easily lead to dramatic consequences in such critical settings. Beyond industrial perspectives, this work could be of general interest for risk-averse decision-making. For instance, parametrized epidemiological models have been used to represent the propagation of Covid-19 and study the impact of lockdown policies. These model parameters are estimated from observational data and corresponding confidence intervals are often available, but rarely used in the decision-making loop. In contrast, our approach would enable evaluating and optimising the worst-case outcome of such public policies.

## Acknowledgments and Disclosure of Funding

This work was supported by the French Ministry of Higher Education and Research, and CPER Nord-Pas de Calais/FEDER DATA Advanced data science and technologies 2015-2020.

## Footnotes

[1]Code and videos available at https://eleurent.github.io/robust-beyond-quadratic/.

[2]We say that a matrix is Metzler when all its non-diagonal coefficients are non-negative.

[3]The expansion of a leaf node $a$ refers to the simulation of its children transitions $aA = \{ab, b \in A\}$

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
