[Supplementary Material]

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

# Supplementary Material

**Outline** In Appendix A, we provide a proof for every novel result introduced in this paper. Appendix B provides additional details on our experiments. Appendix C gives a better method of conversion from ellipsoid to polytope than that of (9). Finally, Appendix D highlights the fact that robustness cannot be recovered by aggregating independent solutions to many optimal control problem.

## A  Proofs

### A.1  Proof of Proposition 1

*Proof.* We differentiate $J(\theta) = \sum_{n=1}^{N} \|y_n - \Phi_n\theta\|^2_{\Sigma_p^{-1}} + \lambda\|\theta\|^2$ as in (6) with respect to $\theta$:

$$
\nabla_\theta J(\theta) = \sum_{n=1}^{N} \nabla_\theta (y_n - \Phi_n\theta)^\mathsf{T}\Sigma_p^{-1}(y_n - \Phi_n\theta) + \nabla_\theta \lambda\|\theta\|^2
$$

$$
= -2\sum_{n=1}^{N} y_n^\mathsf{T}\Sigma_p^{-1}\Phi_n + 2\sum_{n=1}^{N} \theta^\mathsf{T}(\Phi_n^\mathsf{T}\Sigma^{-1}\Phi_n) + 2\lambda\theta^\mathsf{T}
$$

Hence,

$$
\nabla_\theta J(\theta) = 0 \iff \left(\sum_{n=1}^{N} \Phi_n^\mathsf{T}\Sigma_p^{-1}\Phi_n + I_d\right)\theta = \sum_{n=1}^{N} y_n^\mathsf{T}\Sigma_p^{-1}\Phi_n
$$

$\square$

### A.2  Proof of Theorem 1

We start by showing a preliminary proposition:

**Proposition 5** (Matrix version of Theorem 1 of Abbasi-Yadkori et al. 2)**.** *Let $\{F_n\}_{n=0}$ be a filtration. Let $\{\eta_n\}_{n=1}^\infty$ be a $\mathbb{R}^p$-valued stochastic process such that $\eta_n$ is $F_n$-measurable and $\mathbb{E}\left[\eta_n \mid F_{n-1}\right]$ is $\Sigma_p$-sub-Gaussian.*

*Let $\{\Phi_n\}_{n=1}^\infty$ be an $\mathbb{R}^{p\times d}$-valued stochastic process such that $\Phi_n$ is $F_n$-measurable. Assume that $G$ is a $d \times d$ positive definite matrix. For any $n \geq 0$, define:*

$$
\overline{G}_n = G + \sum_{s=1}^{n} \Phi_s^\mathsf{T}\Sigma_p^{-1}\Phi_s \in \mathbb{R}^{d\times d} \quad S_n = \sum_{s=1}^{n} \Phi_s^\mathsf{T}\Sigma_p^{-1}\eta_s \in \mathbb{R}^d.
$$

*Then, for any $\delta > 0$, with probability at least $1 - \delta$, for all $n \geq 0$,*

$$
\|S_n\|_{\overline{G}_n^{-1}} \leq \sqrt{2\ln\left(\frac{\det\left(\overline{G}_n\right)^{1/2}}{\delta \det(G)^{1/2}}\right)}.
$$

*Proof.* Let

$$
G_t = \sum_{s=1}^{t} \Phi_s^\mathsf{T}\Sigma_p^{-1}\Phi_s \in \mathbb{R}^{d\times d}
$$

And for any $z \in \mathbb{R}^d$,

$$
M_t^z = \exp\left(\langle z, S_t\rangle - \frac{1}{2}\|z\|_{G_t}\right)
$$

$$
D_t^z = \exp\left(\langle \Phi_t z, \eta_t\rangle_{\Sigma_p^{-1}} - \frac{1}{2}\|\Phi_t z\|_{\Sigma_p^{-1}}\right)
$$

Then,

$$M_t^z = \exp\left(\sum_{s=1}^t z^\top \Phi_s^\top \Sigma_p^{-1} \eta_s - \frac{1}{2}(\Phi_s z)^\top \Sigma_p^{-1}(\Phi_s z)\right)$$

$$= \prod_{s=1}^t D_s^z$$

and using the Sub-Gaussianity of $\eta_t$

$$\mathbb{E}\left[D_t^z \mid F_{t-1}\right] = \exp\left(-\frac{1}{2}\|\Phi_t z\|_{\Sigma_p^{-1}}\right)$$

$$\mathbb{E}\left[\exp\left(\langle \Phi_t z, \eta_t\rangle_{\Sigma_p^{-1}}\right) \,\Big|\, F_{t-1}\right]$$

$$\leq \exp\left(-\frac{1}{2}\|\Phi_t z\|_{\Sigma_p^{-1}}\right)$$

$$\exp\left((z^\top \Phi_t^\top \Sigma_p^{-1})\Sigma_p(\Sigma_p^{-1}\Phi_t z)\right)$$

$$= 1$$

$$\mathbb{E}\left[M_t^z \mid F_{t-1}\right] = \left(\prod_{s=1}^{t-1} D_s^z\right)\mathbb{E}\left[D_t^z \mid F_{t-1}\right] \leq M_{t-1}^z$$

Showing that $(M_t^z)_{t=1}^\infty$ is indeed a supermartingale and in fact $\mathbb{E}[M_t^z] \leq 1$. It then follows by Doob's upcrossing lemma for supermartingale that $M_\infty^z = \lim_{t\to\infty} M_t^z$ is almost surely well-defined, and so is $M_\tau^z$ for any random stopping time $\tau$.

Next, we consider the stopped martingale $M_{\min(\tau,t)}^z$. Since $(M_t^z)_{t=1}^\infty$ is a non-negative supermartingale and $\tau$ is a random stopping time, we deduce by Doob's decomposition that

$$\mathbb{E}[M_{\min(\tau,t)}^z] = \mathbb{E}[M_0^z] + \mathbb{E}[\sum_{s=0}^{t-1}(M_{s+1}^z - M_s^z)\mathbb{I}\{\tau > s\}]$$

$$\leq 1 + \mathbb{E}[\sum_{s=0}^{t-1}\mathbb{E}[M_{s+1}^z - M_s^z|F_s]\mathbb{I}\{\tau > s\}]$$

$$\leq 1$$

Finally, an application of Fatou's lemma show that $\mathbb{E}[M_\tau^z] = \mathbb{E}[\liminf_{t\to\infty} M_{\min(\tau,t)}^z] \leq \liminf_{t\to\infty} \mathbb{E}[M_{\min(\tau,t)}^z] \leq 1$.

This results allows to apply a result from [36]:

**Lemma 2** (Theorem 14.7 of [36]). *If $Z$ is a random vector and $B$ is a symmetric positive definite matrix such that*

$$\forall \gamma \in \mathbb{R}^d, \ln \mathbb{E}\exp\left(\gamma^\top Z - \frac{1}{2}\gamma^\top B\gamma\right) \leq 0,$$

*then for any positive definite non-random matrix C, it holds*

$$\mathbb{E}\left[\sqrt{\frac{\det(C)}{\det(B+C)}} \exp\left(\frac{1}{2}\|Z\|_{(B+C)^{-1}}^2\right)\right] \leq 1.$$

*In particular, by Markov inequality, for all $\delta \in (0,1)$,*

$$\mathbb{P}\left(\|Z\|_{(B+C)^{-1}} \geq \sqrt{2\ln\left(\frac{\det\left((B+C)^{1/2}\right)}{\delta \det(C)^{1/2}}\right)}\right) \leq \delta.$$

Here, by using $Z = \sum_{s=1}^{t} \Phi_s \Sigma_p^{-1} \eta_s$, $B = G_t$, $C = G$,

$$\mathbb{P}\left( \|S_t\|_{(G_t+G)^{-1}} \geq \sqrt{2 \ln \left( \frac{\det(G_t+G)^{1/2}}{\delta \det(G)^{1/2}} \right)} \right) \leq \delta$$

$\square$

Having shown this preliminary result, we move on to the proof of Theorem 1.

*Proof.* For all $x \in \mathbb{R}^d$, (7) gives:

$$x^\mathsf{T}\theta_{N,\lambda} - x^\mathsf{T}\theta = x^\mathsf{T} G_{N,\lambda}^{-1} \sum_{n=1}^{N} \Phi_n^\mathsf{T} \Sigma_p^{-1} \eta_n - \lambda x^\mathsf{T} G_{N,\lambda}^{-1} \theta$$

$$= \langle x, \sum_{n=1}^{N} \Phi_n^\mathsf{T} \Sigma_p^{-1} \eta_n \rangle_{G_{N,\lambda}^{-1}} - \lambda \langle x, \theta \rangle_{G_{N,\lambda}^{-1}}$$

Using the Cauchy-Schwartz inequality, we get:

$$|x^\mathsf{T}\theta_{N,\lambda} - x^\mathsf{T}\theta| \leq \|x\|_{G_{N,\lambda}^{-1}} \left( \left\| \sum_{n=1}^{N} \Phi_n^\mathsf{T} \Sigma_p^{-1} \eta_n \right\|_{G_{N,\lambda}^{-1}} + \lambda \|\theta\|_{G_{N,\lambda}^{-1}} \right)$$

In particular, for $x = G_{N,\lambda}(\theta_{N,\lambda} - \theta)$, we get after simplifying with $\|\theta_{N,\lambda} - \theta\|_{G_{N,\lambda}}$:

$$\|\theta_{N,\lambda} - \theta\|_{G_{N,\lambda}} \leq \left\| \sum_{n=1}^{N} \Phi_n^\mathsf{T} \Sigma_p^{-1} \eta_n \right\|_{G_{N,\lambda}^{-1}} + \lambda \|\theta\|_{G_{N,\lambda}^{-1}}$$

By applying Proposition 5 with $G = \lambda I_d$, we obtain that with probability at least $1 - \delta$,

$$\|\theta_{N,\lambda} - \theta\|_{G_{N,\lambda}} \leq \sqrt{2 \ln \left( \frac{\det(G_{N,\lambda})^{1/2}}{\delta \det(\lambda I_d)^{1/2}} \right)} + \lambda \|\theta\|_{G_{N,\lambda}^{-1}}$$

And since $\|\theta\|_{G_{N,\lambda}^{-1}}^2 \leq 1/\lambda_{\min}(G_{N,\lambda}) \|\theta\|_2^2 \leq 1/\lambda \|\theta\|_2^2$ and $\|\theta\|_2^2 \leq d\|\theta\|_\infty^2 \leq dS^2$,

$$\|\theta_{N,\lambda} - \theta\|_{G_{N,\lambda}} \leq \sqrt{2 \ln \left( \frac{\det(G_{N,\lambda})^{1/2}}{\delta \det(\lambda I_d)^{1/2}} \right)} + (\lambda d)^{1/2} S$$

$\square$

### A.3  Proof of Theorem 2

*Proof.* The predictor designed in Section 3 verifies the inclusion property (4). Thus, for sequence of controls $\mathbf{u}$, any dynamics $A(\theta) \in C_{N,\delta}$, and disturbances $\underline{\omega} \leq \omega \leq \overline{\omega}$, the corresponding state at time $t_n$ is bounded by $\underline{x}_n \leq x_n \leq \overline{x}_n$, which implies that $R(x_n) \geq \min_{x \in [\underline{x}_n(\mathbf{u}), \overline{x}_n(\mathbf{u})]} R(x) = \underline{R}_n(\mathbf{u})$.

Thus, by taking the min over $C_{N,\delta}$ and $[\underline{\omega}, \overline{\omega}]$, we also have for any sequence of controls $\mathbf{u}$:

$$V^r(\mathbf{u}) = \min_{\substack{A(\theta) \in C_{N,\delta} \\ \underline{\omega} \leq \omega \leq \overline{\omega}}} \sum_{n=N+1}^{\infty} \gamma^n R(x_n)$$

$$\geq \sum_{n=N+1}^{\infty} \gamma^n \underline{R}_n(\mathbf{u})$$

$$= \hat{V}^r(\mathbf{u})$$

And $V^r(\mathbf{u}) \leq V(\mathbf{u}) = \mathbb{E}_\omega \left[ \sum_{n=N+1}^{\infty} \gamma^n R(x_n) \right]$ by definition. $\square$

### A.4 Proof of Theorem 3

We first bound the model estimation error.

**Lemma 3.**

$$\|A(\theta) - A(\theta_{N,\lambda})\|_F = \mathcal{O}\left(\sqrt{\frac{\beta_N(\delta)^2}{\lambda_{\min}(G_{N,\lambda})}}\right)$$

*Proof.* We have

$$\|\theta - \theta_{N,\lambda}\|^2_{G_{N,\lambda}} \geq \lambda_{\min}(G_{N,\lambda})\|\theta - \theta_{N,\lambda}\|^2_2$$

And (8) gives

$$\|\theta - \theta_{N,\lambda}\|^2_{G_{N,\lambda}} = \mathcal{O}(\beta_N(\delta)^2)$$

Moreover, $A(\theta)$ belongs to a linear image of this $L^2$-ball. By writing a the $j^{th}$ column of a matrix $M$ as $M_j$, and its coefficient $i, j$ as $M_{i,j}$,

$$((A(\theta) - A(\theta_{N,\lambda}))^\mathsf{T}(A(\theta) - A(\theta_{N,\lambda})))_{i,j} = (\theta - \theta_{N,\lambda})^\mathsf{T}\phi_i^\mathsf{T}\phi_j(\theta - \theta_{N,\lambda})$$
$$\leq \lambda_{\max}(\phi_i^\mathsf{T}\phi_j)\|\theta - \theta_{N,\lambda}\|^2_2$$

Thus, $\|A(\theta) - A(\theta_{N,\lambda})\|^2_F = \text{Tr}\left[(A(\theta) - A(\theta_{N,\lambda}))^\mathsf{T}(A(\theta) - A(\theta_{N,\lambda}))\right] = \mathcal{O}\left(\frac{\beta_N(\delta)^2}{\lambda_{\min}(G_{N,\lambda})}\right)$

$\square$

Then, we propagate this estimation error through the state prediction.

**Lemma 4.** *If there exist $P > 0, Q_0 \in \mathbb{R}^{p\times p}, \rho > 0$ such that*

$$\begin{bmatrix} A_N^\mathsf{T}P + PA_N + Q_0 & P|D| \\ |D|^\mathsf{T}P & -\rho I_r \end{bmatrix} < 0,$$

*then for all $t > t_N$,*

$$\|\overline{x}(t) - \underline{x}(t)\| \leq \left(C_0 + \mathcal{O}\left(\frac{\beta_N(\delta)}{\sqrt{\lambda_{\min}(G_{N,\lambda})}}\right)\right)C_\omega(t),$$

*where*

$$C_0 = \sqrt{\frac{2\rho\lambda_{\max}(P)}{\lambda_{\min}(P)\lambda_{\min}(Q_0)}},$$

*and*

$$C_\omega(t) = \sup_{\tau \in [0,t]} \|\overline{\omega}(\tau) - \underline{\omega}(\tau)\|_2.$$

*Proof.* Let $e = \overline{x} - \underline{x}$. (11) gives the dynamics

$$\dot{e} = A_N e + |\Delta A|(\overline{x}^+ + \underline{x}^-) + |D|(\overline{\omega} - \underline{\omega})$$

where recall that $|M| = M^+ + M^-$ for any matrix $M \in \mathbb{R}^{p\times p}$.

We define the Lyapunov function $V = e^\mathsf{T}Pe$, which is non-negative definite provided that $P > 0$, and compute its derivative

$$\dot{V} = X^\mathsf{T}\begin{bmatrix} A_N^\mathsf{T}P + PA_N + Q & P|D| & P|\Delta A| \\ |D|^\mathsf{T}P & -\rho I_r & 0 \\ |\Delta A|^\mathsf{T}P & 0 & -\alpha I_p \end{bmatrix} X$$
$$- e^\mathsf{T}Qe + \alpha|\underline{x}^+ + \overline{x}^-|^2 + \rho|\overline{\omega} - \underline{\omega}|^2$$

with $X = \begin{bmatrix} e & \overline{\omega} - \underline{\omega} & \underline{x}^+ + \overline{x}^- \end{bmatrix}^\mathsf{T}$, for any $Q \in \mathbb{R}^{p\times p}, \rho, \alpha \in \mathbb{R}$.

Moreover, it holds that $-\underline{x}^+ - \overline{x}^- \le e \le \overline{x}^+ + \underline{x}^-$, which implies $|\overline{x}^+ + \overline{x}^-| \le 2|e|$. Hence,

$$\dot{V} \le X^{\mathsf{T}} \underbrace{\left[ \begin{array}{cc|c} A_N^{\mathsf{T}}P + PA_N + Q + 4\alpha I_p & P|D| & P|\Delta A| \\ |D|^{\mathsf{T}}P & -\rho I_r & 0 \\ \hline |\Delta A|^{\mathsf{T}}P & 0 & -\alpha I_p \end{array} \right]}_{\Upsilon} X$$
$$- e^{\mathsf{T}}Qe + \rho\|\overline{\omega} - \underline{\omega}\|_2^2$$

Thus, if we had $\Upsilon \le 0$, $Q > 0$, $\rho > 0$, then we would have

$$\dot{V} \le -\mu V + \rho\|\overline{\omega} - \underline{\omega}\|_2^2$$

with $\mu = \frac{\lambda_{\min}(Q)}{\lambda_{\max}(P)}$. Since $V(t_N) = 0$, this further implies that for all $t > t_N$,

$$V(t) \le \frac{\rho}{\mu}C_\omega^2(t) \tag{17}$$

We now examine the condition $\Upsilon \le 0$. We resort to its Schur complement: given $\alpha > 0$, $\Upsilon \le 0$ if and only if $R \ge S$, where $S = \alpha^{-1}\left[|\Delta A|^{\mathsf{T}}P \quad 0\right]^{\mathsf{T}}\left[|\Delta A|^{\mathsf{T}}P \quad 0\right]$ and $R$ is the top-left block of $-\Upsilon$:

$$R = \begin{bmatrix} -A_N^{\mathsf{T}}P - PA_N - Q - 4\alpha I_p & -P|D| \\ -|D|^{\mathsf{T}}P & \rho I_r \end{bmatrix}$$

Choose $Q = \frac{1}{2}Q_0 - 4\alpha I_p$. Assume that $P$ is fixed and satisfies the conditions of the lemma. We have

$$\lambda_{\max}(S) \le \alpha^{-1}\lambda_{\max}(P)^2\lambda_{\max}(|\Delta A|^{\top}|\Delta A|)$$
$$\le \alpha^{-1}\lambda_{\max}(P)^2\|\Delta A\|_F^2$$

Thus, by taking $\alpha = \frac{2\lambda_{\max}(P)^2\|\Delta A\|_F^2}{\lambda_{\min}(Q_0)} = \mathcal{O}(\frac{\beta_N(\delta)^2}{\lambda_{\min}(G_{N,\lambda})})$, we can obtain that $S \le \begin{bmatrix} \frac{1}{2}Q_0 & 0 \\ 0 & 0 \end{bmatrix}$. Thus,

$$R - S \ge \begin{bmatrix} -A_N^{\mathsf{T}}P - PA_N - Q_0 & -P|D| \\ -|D|^{\mathsf{T}}P & \rho I_r \end{bmatrix} > 0$$

as it is assumed in the conditions of the lemma. Hence, under such a choice of $\alpha$ and $Q$, we recover $\Upsilon \le 0$. (17) follows with $\mu = \frac{\lambda_{\min}(Q)}{\lambda_{\max}(P)} = \frac{\frac{1}{2}\lambda_{\min}(Q_0) - 4\alpha}{\lambda_{\max}(P)}$. Finally, we obtain

$$\|e(t)\|_2^2 \le \lambda_{\min}(P)^{-1}V(t)$$
$$\le \frac{2\rho\lambda_{\max}(P)/\lambda_{\min}(P)}{\lambda_{\min}(Q_0) - 8\alpha}C_\omega^2(t)$$

Developing at the first order in $\alpha$ gives

$$\|e(t)\|_2 \le C_0\left(1 + \frac{4\alpha}{\lambda_{\min}(Q_0)} + \mathcal{O}(\alpha^2)\right)C_\omega(t)$$
$$\le \left(C_0 + \mathcal{O}\left(\frac{\beta_N(\delta)^2}{\lambda_{\min}(G_{N,\lambda})}\right)\right)C_\omega(t)$$

$\square$

Finally, we propagate the state prediction error bound to the pessimistic rewards and surrogate objective to get our final result.

*Proof.* For any sequence of controls $\mathbf{u}$, dynamical parameters $\theta \in C_{N,\delta}$ and disturbances $\underline{\omega} \le \omega \le \overline{\omega}$, we clearly have

$$V(\mathbf{u})^r \le V(\mathbf{u}) = \mathbb{E}_{\omega} \sum_n \gamma^n R(x_n)$$

Moreover, by the inclusion property (4), we have that $\underline{x}_n \leq x_n \leq \overline{x}_n$, which implies that $R(x_n) \leq \max_{x \in [\underline{x}_n(\mathbf{u}), \overline{x}_n(\mathbf{u})]} R(x)$. Assuming $R$ is $L$-lipschitz,

$$
\begin{aligned}
V(\mathbf{u}) - \hat{V}^r(\mathbf{u}) &\leq \sum_{n=N+1}^{\infty} \gamma^n \left(\max_{x \in [\underline{x}_n(\mathbf{u}), \overline{x}_n(\mathbf{u})]} - \min\right) R(x) \\
&\leq \sum_{n=N+1}^{\infty} \gamma^n L \left\| \underline{x}_n(\mathbf{u}) - \overline{x}_n(\mathbf{u}) \right\|_2 \\
&\leq L \left( C_0 + \mathcal{O}\left( \frac{\beta_N(\delta)^2}{\lambda_{\min}(G_{N,\lambda})} \right) \right) \sum_{n>N} \gamma^n C_\omega(t_n) \\
&= \Delta_\omega + \mathcal{O}\left( \frac{\beta_N(\delta)^2}{\lambda_{\min}(G_{N,\lambda})} \right)
\end{aligned}
$$

with $\Delta_\omega = LC_0 \sum_{n>N} \gamma^n C_\omega(t_n)$, which is finite by Assumption 2.

Finally, we use the result of Lemma 1 to account for planning with a finite budget, and relate $\hat{V}^r(a^\star)$ to $\hat{V}^r(a_K)$. $\qquad \square$

## A.5 Proof of Corollary 1

*Proof.* By (7) and (14), we have

$$
\lambda_{\min}(G_{N,\lambda}) \geq (N - n_0)\underline{\phi}^2 + \sum_{n < n_0} \Phi_n^\mathsf{T} \Sigma_p^{-1} \Phi_n
$$

and by (8),

$$
\begin{aligned}
\beta_N(\delta) &= \sqrt{2 \log \left( \frac{\det(G_{N,\lambda})^{1/2}}{\delta \det(\lambda I_d)^{1/2}} \right)} + (\lambda d)^{1/2} S \\
&\leq \sqrt{\log \left( N^{d/2} \overline{\phi}^d / (\delta \lambda^{d/2}) \right)} + \mathcal{O}(1)
\end{aligned}
$$

Thus,

$$
\frac{\beta_N(\delta)^2}{\lambda_{\min}(G_{N,\lambda})} = \mathcal{O}\left( \frac{\log(N^{d/2}/\delta)}{N} \right)
$$

**Stability condition 2.** By Lemma 3 and the above, the sequence $(A_N)_N$ converges to $A(\theta)$ in Frobenius norm. Thus,

$$
M_n \stackrel{def}{=} \begin{bmatrix} A_N^\mathsf{T} P + P A_N + Q_0 & P|D| \\ |D|^\mathsf{T} P & -\rho I_r \end{bmatrix} \text{ also converges to } M \stackrel{def}{=} \begin{bmatrix} A(\theta)^\mathsf{T} P + P A(\theta) + Q_0 & P|D| \\ |D|^\mathsf{T} P & -\rho I_r \end{bmatrix},
$$

which is assumed to be negative definite.

Moreover, the two functions that map a matrix to its characteristic polynomial and a polynomial to its roots, are both continuous. Thus, by continuity, the largest eigenvalue of $M_n$ converges to that of $M$, which is strictly negative. Hence, there exists some $N_0 \in \mathbb{N}$ such that for all $N > N_0$, $M_N$ is negative definite, as required in the condition 2. of Theorem 3. $\qquad \square$

## A.6 Proof of Proposition 4

We start by showing the following lemma:

**Lemma 5** (Robust values ordering). *In addition to the robust B-value defined in* (16)*, that we extend to inner nodes*

$$
B_a^r(k) \stackrel{def}{=} \begin{cases} \min_{m \in [M]} \sum_{n=0}^{h-1} \gamma^n R_n^m + \frac{\gamma^h}{1-\gamma} & \text{if } a \text{ is a leaf;} \\ \max_{b \in \mathcal{A}} B_{ab}^r(k) & \text{else.} \end{cases} \tag{18}
$$

*we also define the robust value of a sequence of actions* $a$

$$V_a^r \overset{def}{=} \max_{\mathbf{u} \in a\mathcal{A}^\infty} \min_{m \in [M]} \sum_{n=h(a)+1}^{\infty} \gamma^n R_n^m \tag{19}$$

*and the robust U-values of a sequence of action* $a$

$$U_a^r(K) \overset{def}{=} \begin{cases} \min_{m \in [M]} \sum_{n=0}^{h-1} \gamma^n R_n^m & \text{if } a \text{ is a leaf;} \\ \max_{b \in \mathcal{A}} U_{ab}^r(n) & \text{else.} \end{cases} \tag{20}$$

*Then, the robust values, U-values and B-values exhibit similar properties as the optimal values, U-values and B-values, that is: for all* $0 < k < K$ *and* $a \in \mathcal{T}_T$,

$$U_a^r(k) \leq U_a^r(K) \leq V_a^r \leq B_a^r(K) \leq B_a^r(k) \tag{21}$$

*Proof.* By definition, when starting with sequence $a$, the value $U_a^m(k)$ represents the minimum admissible reward, while $B_a^m(k)$ corresponds to the best admissible reward achievable with respect to the the possible continuations of $a$. Thus, for all $a \in \mathcal{A}^*$, $U_a^m(k)$ and $U_a^r(k)$ are non-decreasing functions of $k$ and $B_a^m(k)$ and $B_a^r(k)$ are a non-increasing functions of $k$, while $V_a^m$ and $V_a^r$ do not depend on $k$.

Moreover, since the reward function $R$ is assumed be bounded in $[0, 1]$, the sum of discounted rewards from a node of depth $d$ is at most $\gamma^d + \gamma^{d+1} + \cdots = \frac{\gamma^d}{1-\gamma}$. As a consequence, for all $k \geq 0$ , $a \in \mathcal{L}_k$ of depth $d$, and any sequence of rewards $(R_n)_{n \in \mathbb{N}}$ obtained from following a path in $a\mathcal{A}^\infty$ with any dynamics $m \in [M]$:

$$U_a^m(k) = \sum_{n=0}^{d-1} \gamma^n R_n^m \leq \sum_{n=0}^{\infty} \gamma^n R_n^m \leq \sum_{n=0}^{d-1} \gamma^n R_n^m + \frac{\gamma^d}{1-\gamma} = B_a^m(k)$$

Hence,

$$\min_{m \in [M]} U_a^m(k) \leq \min_{m \in [M]} \sum_{n=0}^{\infty} \gamma^n R_n \leq \min_{m \in [M]} B_a^m(k) \tag{22}$$

And as the left-hand and right-hand sides of (22) are independent of the particular path that was followed in $a\mathcal{A}^\infty$, it also holds for the robust path:

$$\min_{m \in [M]} U_i^m(k) \leq \max_{a' \in a\mathcal{A}^\infty} \min_{m \in [M]} \sum_{t=0}^{\infty} \gamma^n R_n^m \leq \min_{m \in [M]} B_i^m(k)$$

that is,

$$U_a^r(k) \leq V_a^r \leq B_a^r(k) \tag{23}$$

Finally, (23) is extended to the rest of $\mathcal{T}_k$ by recursive application of (19), (20) and (18). $\qquad \square$

We now turn to the proof of the theorem.

*Proof.* Hren & Munos [19] first show in Theorem 2 that the simple regret $r_K$ of their optimistic planner is bounded by $\frac{\gamma^{d_K}}{1-\gamma}$ where $d_K$ is the depth of $\mathcal{T}_K$. This properties relies on the fact that the returned action belongs to the deepest explored branch, which we can show likewise by contradiction using Lemma 5. This yields directly that the returned action $a = i_0$ where $i$ is some node of maximal depth $d_K$ expanded at round $k \leq K$, which by selection rule verifies $B_a^r(k) = B_i^r(k) = \max_{x \in \mathcal{A}} B_x^r(k)$ and:

$$\begin{aligned} V^r - V_a^r = V_{a^\star}^r - V_a^r &\leq B_{a^\star}^r(k) - V_a^r \leq B_a^r(k) - U_a^r(k) \\ &= B_i^r(k) - U_i^r(k) \\ &= \frac{\gamma^{d_K}}{1-\gamma}. \end{aligned}$$

Secondly, they bound the depth $d_K$ of $\mathcal{T}_K$ with respect to $K$. To that end, they show that the expanded nodes always belong to the sub-tree $\mathcal{T}_\infty$ of all the nodes of depth $d$ that are $\frac{\gamma^d}{1-\gamma}$-optimal. Indeed, if a node $i$ of depth $d$ is expanded at round $k$, then $B_i^r(k) \geq B_j^r(k)$ for all $j \in \mathcal{L}_k$ by selection rule, thus the max-backups of (16) up to the root yield $B_i^r(k) = B_\emptyset^r(k)$. Moreover, by Lemma 5 we have that $B_\emptyset^r(k) \geq V_\emptyset^r = V^r$ and so $V_i^r \geq U_i^r(k) = B_i^r(k) - \frac{\gamma^d}{1-\gamma} \geq V^r - \frac{\gamma^d}{1-\gamma}$, thus $i \in \mathcal{T}_\infty$.

Then from the definition of $\kappa$ applied to nodes in $\mathcal{T}_\infty$, there exists $d_0$ and $c$ such that the number $n_d$ of nodes of depth $d \geq d_0$ in $\mathcal{T}_\infty$ is bounded by $c\kappa^d$. As a consequence,

$$K = \sum_{d=0}^{d_K} n_d = n_0 + \sum_{d=d_0+1}^{d_K} n_d \leq n_0 + c\sum_{d=d_0+1}^{d_K} \kappa^d.$$

- If $\kappa > 1$, then $K \leq n_0 + c\kappa^{d_0+1}\frac{\kappa^{d_K-d_0}-1}{\kappa-1}$ and thus $d_K \geq d_0 + \log_\kappa \frac{(K-n_0)(\kappa-1)}{c\kappa^{d_0+1}}$.

  We conclude that $r_K \leq \frac{\gamma^{d_K}}{1-\gamma} = \frac{1}{1-\gamma}\left(\frac{(K-n_0)(\kappa-1)}{c\kappa^{d_0+1}}\right)^{\frac{\log\gamma}{\log\kappa}} = \mathcal{O}\left(K^{-\frac{\log 1/\gamma}{\log\kappa}}\right)$.

- If $\kappa = 1$, then $K \leq n_0 + c(d_K - d_0)$, hence we have $r_K = O\left(\gamma^{Kc}\right)$.

$\qquad\square$

# B  Experimental details

In both experiments, we used $\gamma = 0.9$, $\delta = 0.9$ and a planning budget $K = 100$. The disturbances were sampled uniformly in $[-0.1, 0.1]^r$ while the measurements are Gaussian with covariance $\Sigma_s = 0.1^2 I_s$.

## B.1  Obstacle Avoidance

**States**  The system is described by its position $(p_x, p_y)$ and velocity $(v_x, v_y)$:

$$x = \begin{bmatrix} p_x & p_y & v_x & v_y \end{bmatrix}^\top$$

**Actions**  It is acted upon by means horizontal and vertical forces $u = (u_x, u_y) \in [-1, 1]^2$. We discretise the action space into four constant controls, for each direction:

$$\mathcal{A} = \{(-1, -1), (-1, 1), (1, -1), (1, 1)\}$$

**Reward**  The reward encodes the task of navigating to reach a goal state $x_g$ while avoiding collisions with obstacles:

$$R(x) = \delta(x)/(1 + \|x - x_g\|_2),$$

where $\delta(x)$ is 0 whenever $x$ collides with an obstacle, 1 otherwise.

**Dynamics**  The system dynamics consist in a double integrator, with friction parameters $(\theta_x, \theta_y)$:

$$\begin{bmatrix} \dot{p}_x \\ \dot{p}_y \\ \dot{v}_x \\ \dot{v}_y \end{bmatrix} = \begin{bmatrix} 0 & 0 & 1 & 0 \\ 0 & 0 & 0 & 1 \\ 0 & 0 & -\theta_x & 0 \\ 0 & 0 & 0 & -\theta_y \end{bmatrix} \begin{bmatrix} p_x \\ p_y \\ v_x \\ v_y \end{bmatrix} + \begin{bmatrix} 0 \\ 0 \\ u_x \\ u_y \end{bmatrix}.$$

Note that Assumption 3 is always verified.

**DQN baseline**  In addition to the state, knowledge of the obstacles is encoded in the observation as an angular grid of laser-like distance measurements, as well as the goal location relative to the system position. As a model for the $Q$-function, we used a Multi-Layer Perceptron with two hidden layers of size 100. An $\varepsilon$-greedy strategy was used for exploration.

## B.2  Autonomous Driving

In the following, we describe the structure of the dynamical system $f$ representing the couplings and interactions between several vehicles.

**States**   In addition to the ego-vehicle, the scene contains $V$ other vehicles. Any vehicle $i \in [0, V]$ is represented by its position $(x_i, y_i)$, its forward velocity $v_i$ its heading $\psi_i$. The resulting joint state is the traffic description: $x = (x_i, y_i, v_i, \psi_i)_{i \in [0,V]} \in \mathbb{R}^{4V+4}$.

**Actions**   The ego-vehicle is following a fixed path, and the tasks consists in adapting its velocity by means of three actions $\mathcal{A} = \{\text{faster, constant velocity, slower}\}$. They are achieved by a longitudinal linear controller that tracks the desired velocity $v_0$, as described below in the system dynamics.

**Reward**   The reward function $R$ is the following:

$$R(x) = \begin{cases} 1 & \text{if the ego-vehicle is at full velocity;} \\ 0 & \text{if the ego-vehicle has collided with another vehicle;} \\ 0.5 & \text{else.} \end{cases}$$

**Dynamics**   The kinematics of any vehicle $i \in [V]$ are represented by the Kinematic Bicycle Model:

$$\dot{x}_i = v_i \cos(\psi_i),$$
$$\dot{y}_i = v_i \sin(\psi_i),$$
$$\dot{v}_i = a_i,$$
$$\dot{\psi}_i = \frac{v_i}{l} \sin(\beta_i),$$

where $(x_i, y_i)$ is the vehicle position, $v_i$ is its forward velocity and $\psi_i$ is its heading, $l$ is the vehicle half-length, $a_i$ is the acceleration command and $\beta_i$ is the slip angle at the centre of gravity, used as a steering command.

**Longitudinal dynamics**   Longitudinal behaviour is modelled by a linear controller using three features: a desired velocity, a braking term to drive slower than the front vehicle, and a braking term to respect a safe distance to the front vehicle.

Denoting $f_i$ the index of the front vehicle preceding vehicle $i$, the acceleration command can be presented as follows:

$$a_i = \begin{bmatrix} \theta_{i,1} & \theta_{i,2} & \theta_{i,3} \end{bmatrix} \begin{bmatrix} v_0 - v_i \\ -(v_{f_i} - v_i)^- \\ -(x_{f_i} - x_i - (d_0 + v_i T))^- \end{bmatrix},$$

where $v_0, d_0$ and $T$ respectively denote the speed limit, jam distance and time gap given by traffic rules.

**Lateral dynamics**   The lane $L_i$ with the lateral position $y_{L_i}$ and heading $\psi_{L_i}$ is tracked by a cascade controller of lateral position and heading $\beta_i$, which is selected in a way the closed-loop dynamics take the form:

$$\dot{\psi}_i = \theta_{i,5} \left( \psi_{L_i} + \sin^{-1} \left( \frac{\widetilde{v}_{i,y}}{v_i} \right) - \psi_i \right), \tag{24}$$
$$\widetilde{v}_{i,y} = \theta_{i,4}(y_{L_i} - y_i).$$

We assume that the drivers choose their steering command $\beta_i$ such that (24) is always achieved: $\beta_i = \sin^{-1}(\frac{l}{v_i} \dot{\psi}_i)$.

**LPV formulation**   The system presented so far is non-linear and must be cast into the LPV form. We approximate the non-linearities induced by the trigonometric operators through equilibrium linearisation around $y_i = y_{L_i}$ and $\psi_i = \psi_{L_i}$.

This yields the following longitudinal dynamics:

$$\dot{x}_i = v_i,$$
$$\dot{v}_i = \theta_{i,1}(v_0 - v_i) + \theta_{i,2}(v_{f_i} - v_i) + \theta_{i,3}(x_{f_i} - x_i - d_0 - v_i T),$$

where $\theta_{i,2}$ and $\theta_{i,3}$ are set to $0$ whenever the corresponding features are not active.

It can be rewritten in the form
$$\dot{X} = A(\theta)(X - X_c) + \omega.$$
For example, in the case of two vehicles only:

$$X = \begin{bmatrix} x_i \\ x_{f_i} \\ v_i \\ v_{f_i} \end{bmatrix}, \quad X_c = \begin{bmatrix} -d_0 - v_0 T \\ 0 \\ v_0 \\ v_0 \end{bmatrix}, \quad \omega = \begin{bmatrix} v_0 \\ v_0 \\ 0 \\ 0 \end{bmatrix}$$

$$A(\theta) = \begin{array}{c} i \\ f_i \\ i \\ f_i \end{array} \begin{bmatrix} \overset{i}{0} & \overset{f_i}{0} & \overset{i}{1} & \overset{f_i}{0} \\ 0 & 0 & 0 & 1 \\ -\theta_{i,3} & \theta_{i,3} & -\theta_{i,1} - \theta_{i,2} - \theta_{i,3} & \theta_{i,2} \\ 0 & 0 & 0 & -\theta_{f_i,1} \end{bmatrix}$$

The lateral dynamics are in a similar form:

$$\begin{bmatrix} \dot{y}_i \\ \dot{\psi}_i \end{bmatrix} = \begin{bmatrix} 0 & v_i \\ -\frac{\theta_{i,4}\theta_{i,5}}{v_i} & -\theta_{i,5} \end{bmatrix} \begin{bmatrix} y_i - y_{L_i} \\ \psi_i - \psi_{L_i} \end{bmatrix} + \begin{bmatrix} v_i \psi_{L_i} \\ 0 \end{bmatrix}$$

Here, the dependency in $v_i$ is seen as an uncertain parametric dependency, *i.e.* $\theta_{i,6} = v_i$, with constant bounds assumed for $v_i$ using an overset of the longitudinal interval predictor.

**Change of coordinates** In both cases, the obtained polytope centre $A_N$ is non-Metzler. We use the similarity transformation of coordinates of Efimov et al. [15]. Precisely, we choose $\Theta$ such that for any $\theta \in \Theta$, $A(\theta)$ is always diagonalisable with real eigenvalues, and perform an eigendecomposition to compute its change of basis matrix $Z$. The transformed system $X' = Z^{-1}(X - X_c)$ verifies (2) with $A_N$ Metlzer as required to apply the interval predictor of Proposition 3. Finally, the obtained predictor is transformed back to the original coordinates $Z$ by using the following lemma:

**Lemma 6** (Interval arithmetic of Efimov et al. 14). *Let $x \in \mathbb{R}^n$ be a vector variable, $\underline{x} \le x \le \overline{x}$ for some $\underline{x}, \overline{x} \in \mathbb{R}^n$.*

1. *If $A \in \mathbb{R}^{m \times n}$ is a constant matrix, then*
$$A^+ \underline{x} - A^- \overline{x} \le Ax \le A^+ \overline{x} - A^- \underline{x}. \tag{25}$$

2. *If $A \in \mathbb{R}^{m \times n}$ is a matrix variable and $\underline{A} \le A \le \overline{A}$ for some $\underline{A}, \overline{A} \in \mathbb{R}^{m \times n}$, then*
$$\underline{A}^+ \underline{x}^+ - \overline{A}^+ \underline{x}^- - \underline{A}^- \overline{x}^+ + \overline{A}^- \overline{x}^- \le Ax \tag{26}$$
$$\le \overline{A}^+ \overline{x}^+ - \underline{A}^+ \overline{x}^- - \overline{A}^- \underline{x}^+ + \underline{A}^- \underline{x}^-.$$

**DQN baseline** In order to avoid discontinuities in the vehicles headings, the state is encoded as $x = (x_i, y_i, v_i^x, v_i^y, \cos\psi_i, \sin\psi_i)_{i \in [0,V]} \in \mathbb{R}^{6V+6}$, with the ego-vehicle always in the first position. As a model for the $Q$-function, we used the Social Attention architecture from [26], that allows to support an arbitrary number of vehicles as input and enforce an invariance to their order.

## C  A tighter conversion from ellipsoid to polytope

**Lemma 7** (Confidence polytope). *We can enclose the confidence ellipsoid obtained in* (8) *within a polytope $C_\delta$:*

$$\mathcal{C}_\delta = \left\{ A_1 + \sum_{i=1}^{2^d} \lambda_i \Delta A_i : \lambda \in [0,1]^{2^d}, \sum_{i=1}^{2^d} \lambda_i = 1 \right\}. \tag{27}$$

*with*

$$h_k \text{ is the } k^{th} \text{ element of } \{-1, 1\}^d \text{ for } k \in [2^d],$$
$$G_{N,\lambda} = PDP^{-1}, \quad \Delta\theta_k = \beta_N(\delta)P^{-1}D^{-1/2}h_k,$$
$$A_N = A(\theta_{N,\lambda}), \quad \Delta A_k = \Delta\theta_k^\mathsf{T}\Phi.$$

*Proof.* The ellipsoid in (8) is described by:

$$\theta \in \mathcal{C}_\delta \implies (\theta - \theta_{N,\lambda})^\mathsf{T} G_{N,\lambda}(\theta - \theta_{N,\lambda}) \le \beta_N(\delta)^2$$
$$\implies (\theta' - \theta'_{N,\lambda})^\mathsf{T} D(\theta' - \theta'_{N,\lambda}) \le \beta_N(\delta)^2$$
$$\implies \sum_{i=1}^{d} D_{i,i}(\theta'_i - \theta'_{N,\lambda,i})^2 \le \beta_N(\delta)^2$$
$$\implies \forall i, |\theta'_i - \theta'_{N,\lambda,i}| \le D_{i,i}^{-1/2}\beta_N(\delta)$$

This describes a $\mathbb{R}^d$ box containing $\theta' = P\theta$, whose $k^{\text{th}}$ vertex is represented by $\theta'_{N,\lambda} + \beta_N(\delta)D^{-1/2}h_k$. We obtain the corresponding box on $\theta$ by transforming each vertex of the box with $P^{-1}$. □

## D  On the ordering of min and max

In the definition of $B^r_a(k)$ (18) and $U^r_a(k)$ (20) it is essential that the minimum over the models is only taken at the end of trajectories, in the same way as for the robust objective (15) in which the worst-case dynamics is only determined after the action sequence has been fully specified. Assume that $U^r_a(k)$ is instead naively defined as:

$$U^r_a(k) = \min_{m \in [1,M]} U^m_a(k),$$

This would not recover the robust policy, as we show in Figure 6 with a simple counter-example.

Figure 6: From left to right: two simple models and corresponding u-values with optimal sequences in blue; the naive version of the robust values returns sub-optimal paths in red; our robust U-value properly recovers the robust policy in green.