[Reviews · NeurIPS 2020]

Review 1

Summary and Contributions: The authors synthesize a variety of results to construct an end-to-end regret analysis for model predictive control with non-convex costs. They use this analysis to craft a robust objective (and, subsequently, a tractable-to-actually-optimize surrogate objective) and, use demonstrate this algorithm on a nontrivial traffic-planning tasks.

Strengths: Soundness (theoretical grounding, empirical evaluation): The authors do a tremendous job theoretically grounding their algorithm. Additionally, the empirical evaluations clearly demonstrate the superiority of the author’s method (and how the author’s method actually recapitulates the theoretical guarantees of the proposed algorithm). Significance and novelty of the contribution: This work is truly a tour de force in combining several different methods in one package to facilitate robust control on real tasks that people actually care about. It appears to be the first end-to-end result guaranteeing that the best result achieved during planning is achievable by underlying system, which seems quite significant (though I am admittedly not an expert in robust control). Relevance to the NeurIPS community: Devising efficient robust control algorithms is crucial for reinforcement learning to see more widespread use in industry. This work is extremely relevant to the NeurIPS community.

Weaknesses: Soundness (theoretical grounding, empirical evaluation): I think the provided evidence is already quite overwhelming, but I’m always eager to see methods applied to more varied environments. Significance and novelty of the contribution: No obvious weaknesses in significance or novelty. Relevance to the NeurIPS community: Clearly relevant.

Correctness: To my knowledge, yes.

Clarity: The paper is exceptionally clear (barring some extremely minor spellings mistakes).

Relation to Prior Work: The work is extremely well situated with respect to prior work.

Reproducibility: Yes

Additional Feedback: Post author rebuttal edit: I look forward to this analysis being applied to even higher dimensional systems :) I'm keeping my score as is, and am satisfied with the author's response.


Review 2

Summary and Contributions: Update: Thanks for the thoughtful response. Your proposed modifications largely address my concerns. Specifically: - I see now that Thm 2 handles both the discretization of actions and the finite planning horizon, separately from Thm 3. I'm satisfied by your proposed reformulation of Thm 3 to improve clarity. - It is true that the PE assumption is common in classic controls literature, even if it isn't in the more recent online learning literature. Given the scope of the planning problem you consider, I'm satisfied by the proposed addition of a corollary, which will more clearly highlight where the assumption comes into play. - The bounded perturbation assumption is common enough in robust planning that I think its a satisfactory way around the noise process issue. The scope of the proposed changes is, in my view, borderline in terms of whether it necessitates another round of review to assess. But overall I land on the positive side, and have updated my score to reflect this. ------- This paper proposes an MPC algorithm for robust and adaptive control of linear systems with structured uncertainties and nonconvex rewards by combining: asymptotic confidence bounds for linear regression, interval prediction bounds, and tree-based planning over pessimistic rewards. The suboptimality of the proposed approach is bounded, and additionally a data-driven model rejection method is proposed.

Strengths: This work considers a setting relevant to modern control problems which cannot be modelled by simple convex or quadratic cost. It is potentially significant to bridge ideas from modern control and planning with those of online learning and nonasymptotic statistics.

Weaknesses: This paper is ambitious, bringing together many different ideas. A downside is that each individual idea receives only passing attention, and the complexity of the proposed method obscures some of the core concepts. See the "Clarity" section for further comments. The significance of the main result is hampered by its reliance on an assumption that the feature vectors are persistently exciting. Usually, the main contribution of non-asymptotic analyses of linear regression is to show a condition of this form holds a priori based on system inputs. I also have reservations about the soundness of the "regret" claims. First, it is unclear how the suboptimality result in Theorem 3 can be crisply defined as regret. The connections between the finite planning strategies over stage costs and the infinite horizon performance V is also not clear. Some of the reasoning about probabilistic bounds is over infinite horizons and continuous signals is fishy. See the "Correctness" section for further comments.

Correctness: The remark on line 135 suggests that omega can be bounded with high probability by union bounding over n(n+1) events. It is not obvious to me why such a finite set is sufficient. The main result in Theorem 3 focuses on the infinite horizon outcomes encoded by V, but the proposed algorithm plans on finite horizons. The precise connection between the two is not made obvious. Should there be an additional term in the sub-optimality bound to account for the finite planning horizon?

Clarity: There are several ways that clarity could be improved: - The connections between the discrete measurement noise and the continuous time noise process is not clear - The switch to action notation in Section 4 is abrupt, even though it is introduced earlier. It would be good to explicitly remind the reader. - V_u in theorem 3 is not defined in the body of the paper. - Many different topics are addressed in this work; it may be better to cut certain topics to devote more explanation to others. For instance, - the state prediction method in Prop 2 is not used, so there is no need to devote space to it - the multi-model selection setting is disconnected from the rest of the paper, and maybe isn't necessary to include (or it should be included as a motivation from the beginning if the framework can be view in a unified way)

Relation to Prior Work: Yes, related work is adequately explored.

Reproducibility: Yes

Additional Feedback:


Review 3

Summary and Contributions: The paper proposes a method for robust adaptive model predictive control with general, non-quadratic costs. It can provide performance guarantees, and a regret bound. The method is verified empirically on two test systems in simulation.

Strengths: I think this is a very solid paper that explores a question of high practical significance: how to learn a system model to be used for model predictive control, without allowing catastrophic performance in the process, that is, with robustness guarantees. Tools from multiple areas are brought together, and competently integrated to provide an entire solution for the closely relates problems of estimation, prediction, and control. The method might appeal to practitioners in industrial settings, where safety is important, and general exploration strategies common in RL might not be suitable. The paper is probably closer to work in the control systems community, but NeurIPS researchers working on RL might appreciate the different outlook on safety during exploration.

Weaknesses: The dynamics are still assumed to be linear, with the uncertainty parameterized as a linear combination of known feature vectors. This is quite restrictive (but also applicable to many practical problems). The discretization of controls might result in some loss of performance, too.

Correctness: I believe the empirical methodology is correct. I could not verify the correctness of the mathematical details in complete detail.

Clarity: The paper is very well written.

Relation to Prior Work: There is a clear discussion of prior related work, and the contributions of this paper with respect to it.

Reproducibility: Yes

Additional Feedback: Minor typos: P.2, L.71: "a tracking a subgoal" -> "tracking a subgoal" P.3, L.83: "forth" -> "fourth" P.7, L:244: "an medium" -> "a medium" --------------------------------------------------------------------------------------- I have read and taken into account the rebuttal.My score has not changed.


Review 4

Summary and Contributions: Post-rebuttal: I would like to thank the authors for their response. As stated in the original review, I think comparing to DQN will improve the paper. I'm still in favor of acceptance. =================================================== This paper address the problem of robust control of continuous dynamic systems, where the system’s dynamics is unknown but assumed to have a linear structure, with external polytopic disturbance. The proposed approach consists of several steps for each action, first model and confidence region estimation (or refinement), then worst case reward extraction and state estimation bounds, a conservative planning step based on the reward and state bounds, finally one step execution, and repeating the process in an MPC like manner. The paper presents an end to end approach to the robust control problem for unknown dynamics (only the system dynamic matrix is unknown) in an adaptive manner. The paper strives to address problems beyond stability or tracking where quadratic costs are common and well studied, such as obstacle avoidance. The control is obtained with a tree search algorithm, and for some more restrictive cases (which are common in the control literature) the regret of the algorithm is provided. The contributions of the paper are mainly for each of the algorithm steps, and putting everything together in an end to end approach with restrictive regret analysis: – An extension to previous work in confidence regions to an elaborate representation of feature matrices. – An MPC like algorithm solving the worst case value function under polytopic disturbances, for the purpose of state estimation. – Proposing a surrogate value function to encapsulate the worst case rewards on an interval, for the robust control computation. – Extending the proposed approach the multi-model case.

Strengths: The paper is well written, and the approach is described for end to end with regret and regions bounds. Finally the effectiveness is demonstrated on two evaluation examples.

Weaknesses: Evaluation against other methods was not presented. This would improve the paper greatly.

Correctness: Everything seems to be correct.

Clarity: The paper is written clearly and each step is explained.

Relation to Prior Work: The relation to previous work is clearly discussed.

Reproducibility: Yes

Additional Feedback: The Adaptive control community have dealt with similar problems (usually in the context of stability), and came up with solutions, sometimes with resemblance to the ones in this paper. It would be appropriate to give a quick survey of the classical system identification and robust adaptive MPC work that have been done, for example (there are many more) "Robust Adaptive MPC for Systems with Exogeneous Disturbances" by V. Adetola and M. Guay.

[Author Response · NeurIPS 2020]

We thank all reviewers for their careful reading, their insightful and constructive comments, and for recognizing the
importance of this research area (R1,R2,R3,R4), the effort we put in bringing together tools from different areas
(R1,R2,R3,R4), and the soundness of theoretical results (R1,R4). We answer the main concerns below, but will
incorporate all feedback (typos, presentation, additional references) in the final version.

**(R2) Unclear connections between the finite planning and the infinite horizon performance $V$.** We decided to
study separately the errors stemming from the approximation of $V^r$ by $\hat{V}^r$ for any controls $\mathbf{u}$ (in Thm. 3); and those
caused by the optimization of $\hat{V}^r$ with a finite budget $K$ (in Thm. 2). Following your suggestion, we propose to
reformulate Thm. 3 to also account for the effect of finite planning, through the planned action $a_K$ (see next point).

**(R2) Reliance of the main result on the PE assumption.** We resorted to this assumption to get asymptotic near-
optimal performance when $N \to \infty$. It is quite common in the control literature, as recognized by R4. However, as you
pointed out, our non-asymptotic analysis actually yields a stronger input-dependent result. Accordingly, we propose to
remove the PE assumption from Thm. 3, which reverts the $\log N / N$ term in the bound back to the input-dependent
term $\beta_N(\delta)^2 / \lambda_{\min}(G_{N,\lambda})$. Thus, our main result becomes:

$$\hat{V}^r(a_K) \leq V^r(a^\star) \leq V(a^\star) \leq \hat{V}^r(a_K) + \underbrace{\Delta_\omega}_{\substack{\text{robustness to} \\ \text{perturbations}}} + \mathcal{O}\left(\underbrace{\frac{\beta_N(\delta)^2}{\lambda_{\min}(G_{N,\lambda})}}_{\text{estimation error}}\right) + \mathcal{O}\left(\underbrace{K^{-\frac{\log 1/\gamma}{\log \kappa}}}_{\text{planning error}}\right)$$

Then, we will mention in a corollary that under the (restrictive) PE assumption, this estimation error term reduces to the
more explicit form $\mathcal{O}\left(\frac{\log(N^{d/2}/\delta)}{N}\right)$ which ensures asymptotic near-optimality when $N \to \infty$ (and $K \to \infty$).

**(R2) Unclear connection between the discrete measurement noise and the continuous time noise process.** In-
deed, the reasoning of the remark on line 135 is insufficient, since it only bounds $\omega(t)$ at discrete time steps $(t_n)_{n\in\mathbb{N}}$
while the interval prediction requires bounds $\underline{\omega}(t) \leq \omega(t) \leq \overline{\omega}(t)$ that hold at all times $t \geq 0$. We overlooked it, and
are very grateful to R2 for pointing it out. Thankfully, this gap can easily be fixed. The most straightforward way is to
directly assume bounded perturbations, such a hypothesis is both realistic and ubiquitous in the robust MPC literature
[17, 4, 6, 28, 22, 29]. Another solution, in the spirit of our remark, is to further specify the continuous-time noise
process $\omega(t)$: in place of Assumpt. 2, the disturbance $\omega(t)$ can be modeled as a Wiener process $W_t$. This has two effects:
first, the increments $W_{t+\mathrm{d}t} - W_t$ are independently and normally distributed, which allows keeping the estimation
procedure of Thm. 1 by simply replacing $y_n$ and $\Phi_n$ by the differences $\widetilde{y}_n = y_n - y_{n-1}$ and $\widetilde{\Phi}_n = \Phi_n - \Phi_{n-1}$ in the
expressions of $\theta_{N,\lambda}$ and $G_{N,\lambda}$. Second, the running maximum $\max_{0 \leq s \leq t} W_s$ of a Wiener process is Half-normally
distributed, which allows to define bounds $\overline{\omega}_n$ on $W_t$ that hold with probability $\delta_n$ *for all* $0 \leq t \leq t_n$. Then, a union
bound over all $[0, t_n]$ intervals can be applied to effectively bound $\omega(t)$ for all $t \geq 0$. That being said, we would rather
include the first solution (bounded perturbations assumption) in the final version for simplicity.

**(R2) Suboptimality result defined as regret.** The term *simple* (as opposed to cumulative) *regret* has been used in
the planning literature [e.g. 19] as a synonym for *suboptimality*. We agree to use the latter, less ambiguous, instead.

**(R2,R3) Relevance of the multi-model extension.** We added this section to mitigate the assumption of linear
dynamics and uncertainty, which R3 finds quite restrictive. Allowing multiple models enables to make multi-modal
predictions, which we consider a substantial improvement: it was *e.g.* necessary for our simulated driving experiment.

**(R3) The discretization of controls might result in some loss of performance.** Indeed. However, as suggested in
line 67, this issue could be circumvented by resorting to more complex planning algorithms for continuous actions,
such as SOPC [10] which also comes with a regret bound. We settled for discrete actions for ease of presentation.

**(R4) Comparison against other methods.** We have wondered about what methods to compare against. In our related
work, we made the case that three major families of robust control (robust stabilization, robust constraint satisfaction, and
minimax LQ) cannot be applied to our environments, which are not stabilization tasks, involve non-convex constraints
and cannot be solved with quadratic costs. Turning to non-robust RL methods, we argued in lines 249-252 that a
model-based approach with the same dynamics priors and planning algorithm (our *nominal* baseline) would make the
fairest comparison. We have since run an additional comparison to DQN, which converged in 30k samples –causing 2k
collisions– to a policy that still suffers a 6% collision rate. This discussion will be included in the final version.

**(R1) More varied environments.** We share your enthusiasm about trying more applications. Note however that
the driving experiment with a state space of dimension 44 is already quite ambitious compared to usual experimental
settings in similar works, often limited to systems of dimension 2, 3 or 4 [3,4,11,12,17,28,29,34,38].

[Meta-Review · NeurIPS 2020]

It is always great if we can remove a common, but limiting, assumption (in this case, that costs are quadratic). The reviewers were impressed by this paper, and the author response helped clear up some lingering uncertainties, and we are happy to recommend acceptance of this paper.